# Effects of a 12-Week Detraining Period on Physical Capacity, Power and Speed in Elite Swimmers

**DOI:** 10.3390/ijerph19084594

**Published:** 2022-04-11

**Authors:** Wojciech Głyk, Maciej Hołub, Jakub Karpiński, Wojciech Rejdych, Wojciech Sadowski, Alina Trybus, Jakub Baron, Łukasz Rydzik, Tadeusz Ambroży, Arkadiusz Stanula

**Affiliations:** 1Institute of Sport Sciences, Jerzy Kukuczka Academy of Physical Education, Mikołowska 72a, 40-065 Katowice, Poland; w.glyk@awf.katowice.pl (W.G.); m.holub@awf.katowice.pl (M.H.); j.karpinski@awf.katowice.pl (J.K.); w.rejdych@awf.katowice.pl (W.R.); w.sadowski@awf.katowice.pl (W.S.); a.trybus@awf.katowice.pl (A.T.); j.baron@awf.katowice.pl (J.B.); a.stanula@awf.katowice.pl (A.S.); 2Institute of Sport Sciences, University of Physical Education in Krakow, 31-571 Kraków, Poland; tadek@ambrozy.pl

**Keywords:** swim, power, speed, body composition

## Abstract

This study aims to evaluate the effects of a prolonged transition period (detraining) on the physical capacity, power, and speed parameters of elite swimmers. Fourteen swimmers (seven females and seven males) aged 20.4 ± 1.7 years participated in the study. The athletes were subjected to two rounds of identical tests at 12-week intervals during the detraining period (DP), which consisted of an evaluation of the athletes’ body weight and composition, a measurement of the power of their lower limbs (Keiser squat, countermovement jump (CMJ), akimbo countermovement jump (ACMJ)) and upper limbs (Keiser arms) on land, and 20-m swimming using the legs only (Crawl Legs test), arms only (Crawl Arms test), and full stroke (Front Crawl test). An analysis of variance revealed a significant effect of the main factor, Gender, on all the measured parameters, while for the factor Detraining, except for Front Crawl (W) (F = 4.27, *p* = 0.061), no significant interaction effect (Gender × Detraining) was revealed. Among both the male and the female participants, a reduction in lactate-threshold swimming speed (LT Dmax) and a reduction in swimming speed and power on the Crawl Arms, Crawl Legs, and Front Crawl tests was observed after 12 weeks. There were also statistically significant reductions in ACMJ and CMJ jump height and upper-limb power (Keiser squat) among the female and male swimmers. There were no significant changes in body weight or body composition. The study showed a clear deterioration in results for most of the parameters, both for those measured on land and for those measured in water.

## 1. Introduction

In all sports, including swimming, it takes much longer to return to high performance than to lose it, so any breaks in the training process should be planned carefully [1]. The detraining period (DP) is as necessary for every training process as other periods because athletes need to rebuild motivation and the right mental attitudes for a new training cycle [2]. 

The consequences of typical breaks between seasons (with their duration defined as ca. 4 to 6 weeks for elite athletes) are not well defined and may be related to the strategy adopted during the recovery period and the baseline level of fitness [3,4]. However, some of the changes occurring in the cardiorespiratory, neuromuscular, and metabolic systems have been characterized [5]. There are studies in the literature showing both a significant decrease in VO_2_max in well-trained athletes who stopped training for 3–6 weeks [6,7] and a reduction in capillary density, oxidative capacity [5], mean cross-sectional area of muscle fibers [8], EMG activity, and changes in fiber type [9]. According to Kuipers and Keizer, the complete cessation of training in a well-trained, competitive athlete can result in a syndrome of detraining, relaxation, exercise abstinence, or exercise addiction [10]. Such effects are noticed when training is suddenly interrupted due to injury or other unplanned circumstances. In practice, however, an athlete who makes good use of the period of active recovery during the transition period can easily develop his or her physical abilities in the next phase of preparation.

The duration of the break may vary depending on individual coaches’ arrangements, schedules, and the sporting calendars of individual national swimming federations [11]. According to some authors, the transition phase should last from 2 to 4 weeks but can be extended to 6 weeks [2]. Taking into account the calendar established by the Polish Swimming Federation, there are two transition periods: a winter period lasting about two weeks between the Polish Winter Championships and the New Year, and a longer summer period that begins after the Polish National Championships. The latter period can be as long as 3 months, depending on the opportunities to compete after major competitions and the availability of swimming pools, which are usually overhauled during the holiday season. All the swimmers participating in the study are of high national sport level, with their starting calendar covering a transition period of approximately 12 weeks, which is similar to the calendars of other swimming federations, such as the NCAA. There are studies showing changes in high-class athletes after about 6 weeks of detraining [12], as well as in amateurs after more than 16 weeks [13,14,15]. Therefore, it seems interesting to examine the effects of a 12-week transition period based on a standardized and controlled effort not exceeding a specific intensity. To the best of our knowledge, this is the first study based on such measurements on land and in water to show the effects of detraining on a group of adult elite swimmers after a 3-month break from regular training.

The study aimed to determine the magnitude of the changes in the test scores of elite athletes participating in competitive swimming. The study took into account both changes in body composition, such as body weight, muscle mass, percentage of body fat, and the power of the athletes’ arms and legs. In addition, specific tests in water were carried out: a Sprint 1080 device was used to measure the time and power generated during swimming, and the 8 × 100-m test was used to determine the changes in the lactate threshold obtained during swimming, both immediately after the end of the main preparation macrocycle and after a 12-week transition period.

## 2. Materials and Methods

### 2.1. Participants

Nineteen competitive swimmers (8 females and 11 males) entered the study. However, due to the incomplete results of several competitors, the results of 14 swimmers (7 females and 7 males) who participated in all tests in both experimental series were used for the final analysis. The participants’ mean age was 20.4 ± 1.7 years, body height was 179.5 ± 8.8 cm, and body weight was 72.2 ± 9.6 kg at the first and 74.6 ± 11.7 kg at the second measurement performed after the 12-week break. The athletes were characterized by at least 10 years of training experience and were competing in national-level competitions. Their sports scores were 755.3 ± 35.9 points in the FINA classification. All athletes trained in the same training group and were members of the same club. All swimmers participating in the study were healthy during both the first and the second measurement. The protocol and possible risks of the study were explained to all participants before they gave their voluntary and informed written consent. The experiment was approved by the Bioethics Committee of the Jerzy Kukuczka Academy of Physical Education in Katowice (No. 8/2018).

### 2.2. Testing Procedures

Two identical measurements were performed on the same group of 14 swimmers. The first measurement was conducted at the end of the training period (TP), after the athletes’ return from the Polish National Swimming Championships (on 27 May), and before their participation in the Polish Academic Championships, during which the athletes maintained peak in-season performance. The second measurement was performed after 12 weeks of a detraining period (DP) (on 31 August). The length of the transition (detraining) period was due to the arrangement of the sporting calendar and the availability of the training facility.

### 2.3. Body Weight and Composition

Each participant underwent body composition analysis using the electrical bioimpedance method (Tanita BC-420, Tokyo, Japan). The day before the test, none of the tested swimmers consumed alcohol or performed any intensive exercise; the participants also underwent a 2-h minimum fast, had a dry body and emptied their bladders [16]. Before each measurement, information about the participants’ sex, age, body height, and body type was entered using the “Athlete” profile for individuals who participate in sports regularly. Body weight (kg), body fat (%), and muscle mass (kg) were measured.

The measurements in water took place in a 25-m swimming pool of the Academy of Physical Education in Katowice. Identical conditions prevailed during each experimental series, which took place at the same time of day, i.e., air temperature ~25.5 °C, water temperature ~27.5 °C, relative humidity ~60%, and water pH ~ 6.93.

### 2.4. 8 × 100-m Test

The test was conducted based on the procedure outlined by Maglischo [17]. The first series of 3 × 100 m featured 1-min breaks between sections, 75% of maximum effort, 3-min rest after the series, and blood sample between 2 and 3 min. The second series of 2 × 100 m featured a 1-min break between sections, 85% of maximum effort, 4-min rest after the series, and blood sample between 3 and 4 min. The third series of 1 × 100 m featured 90% of maximal effort, with a 6-min break, and blood sample between 4 and 5 min. The fourth series of 1 × 100 m featured 95% of maximal effort, with a 20-min break, and blood sample between 5 and 6 min. The fifth series of 1 × 100 m, at 100% maximum effort, featured a blood sample between 5 and 6 min. Capillary tubes and EKF Diagnostic test tubes were used for blood collection, after which blood samples were analyzed in an EKF Diagnostic Biosen C-line lactate analyzer. To determine the swimming speed at the anaerobic threshold before and after detraining, the Dmax method was used, defined as the maximum perpendicular distance of the lactate curve from the line connecting the start- to the end-point of the lactate curve [18,19].

### 2.5. Measuring Swimming Power and Time

A Sprint 1080 Motion measurement system (1080 Sprint; 1080 Motion, Lidingö, Sweden) was used for the measurements. According to the manufacturer’s data, the system is characterized by high repeatability and accuracy of position (±0.5%), velocity (±0.5%), force (±4.8 N), power, and time measurements [20]. Of all the data recorded by the system over a selected distance of 20 m, the mean values of power (W) and time (s) obtained while covering the entire measured distance were selected. The data were collected into the manufacturer’s compatible software at a frequency of 333 Hz. An individual profile including age, gender, body height, and body weight was created for each study participant for measurement purposes. Each swimmer took part in three tests (Crawl Legs, Crawl Arms, and Front Crawl) over a distance of 20 m with maximum effort and with 1 kg resistance. After each test, the participants rested for ca. 10 min. During the test, the athletes were strapped to the device using a belt placed on their hips in such a way as to not interfere with the measurement. To isolate the work of the legs, a large board held in front of the body was used, while when working with the arms, a pullbuoy was placed between the thighs, with a rubber on the ankles, preventing the use of the legs.

### 2.6. Upper- and Lower-Limb Power on Keiser Pneumatic Resistance System

Each swimmer tested with an individually selected load performed two tests that measured lower-limb and upper-limb power. The tests were preceded by a warm-up, with the same procedure used during both measurements. The lower-limb power test was performed on a Keiser squat system and consisted in performing as many squats as possible with an individual load, depending on the participant’s body weight. The load was verified after the second body weight measurement, performed after 12 weeks of detraining. The test ended when the device recorded a decrease in the athlete’s power output. For the purpose of the study, the number of repetitions performed and the maximum power (Keiser squat) were recorded and included in the results. The upper-limb power test (Keiser arms) had an analogous pattern to that of the lower limbs. It consisted in pulling two cables with handles while lying forward on an inclined bench set at an angle of 30 º and placed 1 m from the Keiser functional trainer. The individual load was equivalent to 10% of body weight and was verified after 12 weeks of detraining.

### 2.7. Countermovement Jump

Countermovement jump test (CMJ) with arm swing and akimbo countermovement jump test (ACMJ) with arms placed on hips were performed. The measurements were taken using OptoJump Next system (Microgate, Bolzano, Italy) using the generally accepted measurement protocol for CMJ and ACMJ tests, as described by Hatze [21]. Verified data of each participant’s body weight were entered each time, and the test itself was preceded by a similar warm-up. Jump height data were recorded with an accuracy of 0.1 cm. The power expressed in watts (W) was computed based on the jump height and current body weight using Sayers’ formula [22]: Pmax [W] = 60.7 × jump height (cm) + 45.3 × body mass (kg) − 2055.

### 2.8. Training Protocol:

The entire group of swimmers studied participated in the same 20-week training under the supervision of the same coaches at the same training facility. The training period (TP) was aimed to develop peak performance for the Polish Adult and Youth Swimming Championships. This macrocycle consisted of 10 training sessions per week in water, 6 days per week, and strength training, performed 2–3 times per week. In the first period, covering weeks 1–4, the athletes swam 139.6 km in total and spent 21 h in the gym. In weeks 5–8, these values were 172.3 km and 18 h, respectively, and in weeks 9–12, the participants swam 227.8 km in the pool and spent 22.5 h in the gym. The final pre-race period included weeks 13–18, when athletes covered 180 km in water and exercised 18 h in the gym to reduce training loads for tapering. After returning from the championships, the athletes maintained top form in weeks 19–20 by swimming 120.3 km. The first measurement for all tests was performed in this period.

The detraining period (DP), which began after the Polish Academic Championships and lasted until the athletes returned to the training facility, lasted 12 weeks. The athletes were instructed to refrain from in-water training throughout the transition period and to keep other physical activities at light and moderate intensity, as determined for each participant (<6.0 METS). Additionally, all participants were advised to maintain a diet similar to that of the training period. Adherence to all of the aforementioned guidelines was verified through ongoing contact with the research staff and was considered an eligibility criterion for further study. After returning to the training facility, 14 athletes qualified for retesting, according to the same testing procedures as described above.

### 2.9. Statistical Analyses

The means and standard deviations (SD) were used to represent the average and normal scatter of the values of all variables. The Shapiro–Wilk W test was used to verify the hypothesis of normality of distribution for the analyzed variables. The effects of the transition period were evaluated using a two-way analysis of variance with a repeated-measures design. The first between-group factor was the Gender of the participants with two levels: Female and Male. The repeated-measures factor was the Detraining—timing of the measurements with two levels, i.e., before the transition period (TP) and after the transition period of 12 weeks (DP). The F-test was used to verify the null hypothesis, whereas the multiplecomparisons procedure, which takes into account the statistical significance of the dependent variable, was verified based on the Tukey test, at a significance level set at α = 0.05. Before proceeding with the analysis of variance, the assumptions for its performance were verified, i.e., testing of the assumption of sphericity using Mauchly’s test. If the assumption of sphericity was not met, the solution of correcting the degrees of freedom in the F-test with the Greenhouse–Geisser coefficient (ε G-G) calculated for this purpose was applied in the subsequent stages of the analysis.

To assess within-group differences (Gender) for parameters recorded before and after the introduction of the detraining period, a t-test for dependent samples was used, while the formula: relative difference (∆%) = ((POST-score − PRE-score)/PRE-score) × 100 was used to assess the percentage change in the scores. Furthermore, to determine the magnitude of the difference between these parameters, a standardized effect size (ES) and 95% confidence intervals were calculated using the mean standard deviation calculated from the measurements of the parameters recorded before and after the detraining period (in the denominator) and the difference between these parameters (in the numerator). Effect sizes were scored as: trivial < 0.2; small ≥ 0.2 and <0.6; medium ≥ 0.6 and <1.2; large ≥ 1.2 and <2; and very large ≥ 2 based on the thresholds defined by Hopkins et al. [23].

All statistical computations were performed by means of the STATISTICA software ver. 13 PL (TIBCO Software Inc., Palo Alto, CA, USA, 2017). Effect sizes and confidence intervals were calculated using Jamovi software (The Jamovi Project 2021, version 2.2, Sydney, Australia) and the esci package (effect sizes and confidence intervals). The ggplot2 package in the r-cran environment was also used for graphical representation of the results [24].

## 3. Results

Table 1 shows the body weights, the body compositions, and the results of the on-land tests. The analysis of variance revealed a statistically significant main effect of the Detraining factor for body weight (F = 7.97, *p* = 0.015) and body fat (F = 9.05, *p* = 0.011) and of the Gender factor for body weight (F = 16.67, *p* = 0.002), body fat (F = 16.19, *p* = 0.002), and muscle mass (F = 54.83, *p* < 0.001). However, no significant interaction was revealed for the Detraining × Gender factors for the variables describing body composition and physique. A detailed analysis of the post-detraining results showed increases in body weight of 1.22 kg (1.9%; ES = trivial) and 3.61 kg (4.6%; ES = small) and in body fat of 1.33% (6.8%; small) and 1.87% (19.3%, ES = moderate) in the female and male groups, respectively. However, these changes were not statistically significant. The analysis of the Keiser arms and Keiser squat tests revealed a statistically significant effect of Detraining and Gender. It should be noted that in the group of females, differences in power in both the upper and the lower limbs recorded after the detraining period, amounting to 51.57 W (9.0%) and 62.29 W (6.6%), respectively, were statistically significant (*p* = 0.003; ES = moderate and *p* = 0.013; ES = small). A decrease in power for Keiser arms and Keiser squat was also reported in the males, but the differences were not statistically significant. The analysis of variance for the ACMJ and CMJ jump scores expressed in both centimeters and watts revealed significant effects of the main factors: Gender (group effect) and Detraining (time effect). Both the females and the males in both jumping varieties achieved statistically significantly lower jump heights after the detraining period. In the females, the CMJ deteriorated by an average of 3.27 cm (37.3 ± 1.80 cm vs. 34.0 ± 2.86 cm; *p* = 0.003; ES = Large), while the males scored 4.69 cm lower on average (49.0 ± 3.58 cm vs. 44.4 ± 6.48 cm; *p* = 0.025; ES = moderate). In terms of ACMJ, the females scored less by 3.36 cm (32.7 ± 1.73 cm vs. 29.3 ± 2.94 m; *p* = 0.006; ES = large), while the males deteriorated by an average of 4.27 cm (42.9 ± 5.00 cm vs. 38.7 ± 4.93 cm; *p* = 0.006; ES = moderate). Figure 1 shows the standardized mean differences (effect size) for the on-land test scores recorded before and after the detraining period.

Table 2 shows the results from the measurements performed in water after the TP and DP periods. The analysis of variance revealed no significant interaction effects for Detraining × Gender in any of the tests conducted. For the anaerobic threshold speed results (LT DMax), the analysis of variance revealed a significant main effect of Gender (F = 78.56, *p* < 0.001) and Detraining (F = 5.37, *p* = 0.039). However, the observed insignificant differences in the measurements performed before and after detraining within each gender were not statistically significant. Of all the tests performed in water, the analysis of variance revealed no significant effect of the Detraining factor (F = 4.27, *p* = 0.061) only for Front Crawl (W). For the same variable, no statistically significant difference was observed for the male group in the measurements separated by the detraining period (32.9 ± 2.76 W vs. 28.9 ± 7.72 W; *p* = 0.229; ES = moderate). Furthermore, in the men’s group, the measurements taken before and after the detraining period revealed no statistically significant differences for either time (12.7 ± 0.76 s vs. 12.9 ± 0.67 s; *p* = 0.547; ES = small) or power (28.0 ± 2.43 W vs. 24.7 ± 5.34 W; *p* = 0.143; ES = moderate) generated when swimming using the arms only, as in the front crawl. In all the other in-water test results for swimming with the legs only, the arms only, and using an entire stroke (Front Crawl test) in both groups, the analysis of variance showed a significant effect of the Gender and Detraining factors. Statistically significant increases in time and decreases in power were also found for all the variables discussed in both gender groups. Figure 2 shows the standardized mean differences (effect size) for the in-water test scores recorded before and after the detraining period.

## 4. Discussion

The main aim of this study was to investigate the level of changes in physical capacity, power, and swimming speed in professional swimmers during a 12-week detraining period. For this purpose, selected and standardized on-land and in-water tests were used to show as accurately as possible the changes in such a specific sport as swimming.

Taking into account the somatic characteristics of the participants, no statistically significant differences were found in any parameters in either the male or the female participants. Interestingly, in a study by Ormsbee and Arciero [25], who examined swimmers after an almost identical transition period in terms of exercise intensity and duration, significant differences in both body fat percentage and body weight were reported. Furthermore, Zacca et al. [11] also found no statistically significant difference in body weight, despite the fact that the study group was only assessed after 4 weeks of detraining. However, it is important to note the advanced skill level and experience of the participants in our study group, which certainly influenced the results.

Lactate-threshold swimming speed (LT DMax) is one of the most commonly used indicators of swimming endurance [26,27], with several methods used to calculate it [28]. There is a belief that detraining can cause significant losses in physiological adaptations [3,4]. The results obtained in this study show significant changes in swimming speed after TP and DP only within gender. There is a disproportion between the results of the females, whose threshold speed did not change, and the males, whose swimming speed at the threshold slightly decreased after the period of detraining. Given evidence in the literature stating that swimming speed reflects the overall status of the swimmer in terms of their strength, fitness, and technical proficiency in water, determining the decline in swimming speed is critical to the evaluation of the changes occurring after a detraining period [29]. With this in mind, coaches should focus on maintaining proper coordination and swimming technique during periods of low-intensity training.

Adequate levels of muscle power in swimmers lead to an improved ability to generate propulsive force in water [30,31,32]. Many studies have emphasized the role of muscle strength as a determinant of athletic performance [33,34]. In order to demonstrate the changes observed in the swimmers in our study, Crawl Arms, Crawl Legs, and Front Crawl tests were performed in the aquatic environment, which is natural for the sport of swimming. The results show a significant decrease in power and swimming speed in both the males and the females between the pre- and post-detraining tests. When swimming with the legs only (Crawl Legs), the power generated decreased in both the male and the female participants by as much as 12.2 percent compared to baseline. Even greater differences were observed when increasing the time to swim the distance measured. Here, the differences were 14.1% and 13.0% in the males and females, respectively. While there are no similar tests in the literature, it is believed that both upper- and lower-body strength and CMJ scores are strongly correlated with in-water fitness [35]. A comparison of the above results to the CMJ jump test performed in the gym also revealed significant decreases in both the male and female groups. However, it should be remembered that the results of the power and the height of the participants’ jumps were strongly correlated with the mass and composition of their bodies, where significant differences were observed and which certainly had an impact on the results obtained in this test. Research has shown a significant decrease in CMJ performance in soccer players after 3 months of detraining [36]. Furthermore, the tests performed on the Keiser machine showing the power generated by the lower limbs during the dynamic squats showed significant decreases in generated power values across the group. However, the females had larger decreases in generated power, whereas these were less significant in the men. Nevertheless, some correlation in the decrease in performance both on land and in water can be emphasized, which provides a clue to coaches as to how the detraining period directly affects the performance of the legs in water.

A similar relationship can be observed in the tests performed only with the arms in water and on land. As with the lower limbs, the tests we performed in water in our study have not been reported in the literature. Given the nature of swimming, the methods of evaluating arm power are less developed compared to those used in other sports [37,38]. Significant decreases in power and an increased time to swim 20 m with the arms only were observed. A two-way analysis of variance showed significant differences in both arm swimming parameters, while differences were also observed between the males and the females. The females had greater power decreases and longer swimming times compared to the males. This coincides exactly with the results obtained in the on-land tests, where the arm power decreased significantly in the whole group. Significant decreases in power were also observed in the females on the in-water tests. It should be noted, however, that the detraining period between the tests was 12 weeks. This is a key factor when analyzing the results. Trinity [39] noted that arm power can increase in the first and third weeks after the cessation of training.

A comparison of the data given above to the results obtained from the athletes swimming with the entire stroke reveals an interesting correlation, consisting in a significant increase in the 20-m swimming time in both the male and the female participants. This is corroborated by other studies, which reported increase in swimming time over various distances following a few weeks of detraining [11]. However, the results obtained for the power generated for the same distance are not as clear. Although the females showed a statistically significant decrease in the power generated during swimming, there were no significant differences within the group on the tests after TP and DP. It is also important to remember that swimming with the entire stroke is affected by more than just the sum of the power of the arms and legs. The results of the study show that the swimming performance for the entire stroke is also largely influenced by central stabilization [40], which, as a component of the swimmer’s preparation, can affect the results described above. Since, to our knowledge, this was the first study to examine both power and swimming time, the correlation noted between the aforementioned values seems to be important for coaches planning post-season breaks.

## 5. Conclusions

In the present study, we observed that long detraining periods have a significant effect on the key parameters that determine swimming performance. Differences were demonstrated between males and females for individual tests, which may be important information for coaches in groups with low gender diversity. Differences were also shown for the whole group without division into gender, which is also significant for planning transition periods. The study included on-land tests, but special attention was paid to the parameters obtained in water, as these are key aspects for swimmers. To date, there have been no studies that have explicitly and directly examined the changes occurring in adult elite swimmers similar to those in our study. Swimming coaches should be aware of how their group responds to a long transition period, such as 12 weeks of detraining. In this way, after returning to regular training, they will be able to periodize training to compensate for the deficiencies that occur during periods of absence from training, and thus shorten the process of athletes’ returning to full fitness by using the most effective training methods.

## Figures and Tables

**Figure 1 ijerph-19-04594-f001:**
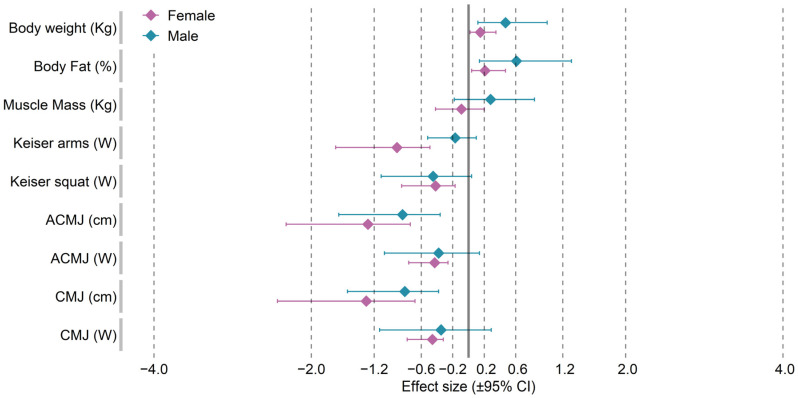
Standardized mean changes and 95% confidence intervals between the pre- and post-detraining evaluation of somatic and physical fitness variables in female and male swimmers.

**Figure 2 ijerph-19-04594-f002:**
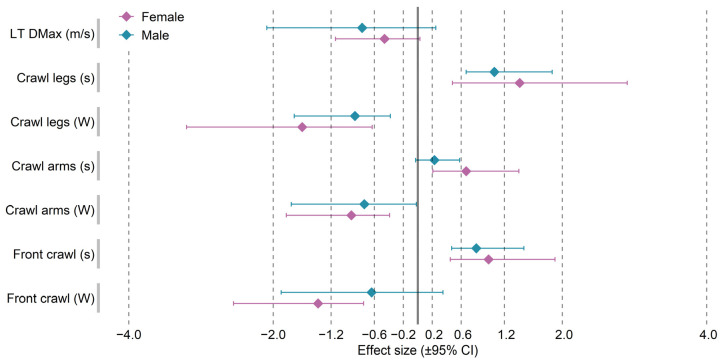
Standardized mean changes and 95% confidence intervals between the pre- and post-detraining evaluation of physical fitness variables in female and male swimmers.

**Table 1 ijerph-19-04594-t001:** Pre- and post-detraining changes in on-land performance. The data are presented as means and standard deviations (±SD).

Variable	Gender	TP	DP	*p*-Value	Change (%)	ANOVA (F, *p*)					
						Detraining (Time Effect)		Gender (Group Effect)		Detraining × Gender	
						F	*p*	F	*p*	F	*p*
Weight (kg)	F	64.9 ± 6.71	66.2 ± 8.05	0.137	−1.22 (−1.9%)	7.97	0.015	16.67	0.002	1.969	0.186
	M	79.4 ± 5.93	83.0 ± 8.39	0.059	−3.61 (−4.6%)						
Body fat (%)	F	19.6 ± 5.17	20.9 ± 6.43	0.107	−1.33 (−6.8%)	9.05	0.011	16.19	0.002	0.260	0.619
	M	9.7 ± 2.20	11.5 ± 3.44	0.058	−1.87 (−19.3%)						
Muscle mass (kg)	F	49.8 ± 4.37	49.3 ± 5.49	0.561	0.50 (1.0%)	0.48	0.500	54.83	<0.001	1.882	0.195
	M	68.1 ± 4.58	69.6 ± 5.72	0.262	−1.53 (−2.2%)						
Keiser arms (W)	F	571 ± 47	519 ± 58	0.003	51.57 (9.0%)	8.18	0.014	28.76	<0.001	0.212	0.654
	M	980 ± 192	942 ± 212	0.246	37.29 (3.8%)						
Keiser squat (W)	F	966 ± 136	903 ± 138	0.013	62.29 (6.6%)	9.81	0.009	37.25	<0.001	0.190	0.671
	M	1465 ± 167	1383 ± 178	0.102	82.43 (5.6%)						
CMJ (cm)	F	37.3 ± 1.80	34.0 ± 2.86	0.003	3.27 (8.8%)	21.48	0.001	30.71	<0.001	0.679	0.426
	M	49.0 ± 3.58	44.4 ± 6.48	0.025	4.69 (9.6%)						
CMJ (W)	F	3150 ± 345	2999 ± 318	<0.001	151.33 (4.8%)	5.11	0.043	71.50	<0.001	0.004	0.952
	M	4583 ± 187	4439 ± 456	0.305	143.35 (3.1%)						
ACMJ (cm)	F	32.7 ± 1.73	29.3 ± 2.94	0.006	3.36 (10.3%)	34.98	<0.001	24.46	<0.001	0.502	0.492
	M	42.9 ± 5.00	38.7 ± 4.93	0.006	4.27 (9.9%)						
ACMJ (W)	F	2873 ± 345	2716 ± 281	0.001	156.54 (5.5%)	5.77	0.033	71.69	<0.001	0.096	0.762
	M	4216 ± 241	4095 ± 389	0.324	120.79 (2.9%)						

Note: TP—training period; DP—detraining period; CMJ—countermovement jump; ACMJ—akimbo countermovement jump.

**Table 2 ijerph-19-04594-t002:** Pre- and post-detraining changes in in-water performance.

Variable	Gender	TP	DP	*p*-Value	Change (%)	ANOVA (F, *p*)					
						Detraining (Time Effect)		Gender (Group Effect)		Detraining × Gender	
						F	*p*	F	*p*	F	*p*
LT DMax (m·s^−1^)	F	1.4 ± 0.07	1.4 ± 0.07	0.096	0.04 (2.5%)	5.37	0.039	78.56	<0.001	0.002	0.965
	M	1.7 ± 0.05	1.6 ± 0.04	0.199	0.04 (2.2%)						
Crawl Legs (s)	F	21.0 ± 1.42	23.6 ± 1.95	0.019	−2.57 (−12.2%)	29.46	<0.001	27.27	<0.001	0.345	0.568
	M	16.9 ± 1.73	18.9 ± 1.91	<0.001	−2.07 (−12.2%)						
Crawl Legs (W)	F	15.9 ± 1.39	13.6 ± 1.26	0.012	2.26 (14.1%)	25.41	<0.001	19.45	0.001	0.257	0.621
	M	21.1 ± 3.35	18.3 ± 2.52	0.011	2.76 (13.0%)						
Crawl Arms (s)	F	15.9 ± 1.25	16.8 ± 1.1	0.023	−0.85 (−5.3%)	6.91	0.022	54.90	<0.001	3.021	0.108
	M	12.7 ± 0.76	12.9 ± 0.67	0.547	−0.17 (−1.3%)						
Crawl Arms (W)	F	21.1 ± 1.95	19.4 ± 1.37	0.014	1.66 (7.9%)	6.07	0.030	20.01	0.001	0.645	0.437
	M	28.0 ± 2.43	24.7 ± 5.34	0.143	3.27 (11.7%)						
Front Crawl (s)	F	13.2 ± 0.62	14.0 ± 0.83	0.009	−0.77 (−5.8%)	21.43	0.001	64.02	<0.001	1.409	0.258
	M	10.9 ± 0.32	11.3 ± 0.67	0.037	−0.46 (−4.2%)						
Front Crawl (W)	F	26.4 ± 1.40	24.2 ± 1.61	0.002	2.23 (8.4%)	4.27	0.061	11.13	0.006	0.337	0.572
	M	32.9 ± 2.76	28.9 ± 7.72	0.229	3.97 (12.0%)						

Note: TP—training period; DP—detraining period; LT—lactate threshold.

## Data Availability

The data presented in this study are available on request from the corresponding author.

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
