# Peer review of "Effects of a 12-Week Detraining Period on Physical Capacity, Power and Speed in Elite Swimmers"

_ijerph, 2022, doi:10.3390/ijerph19084594_

Round 1
Reviewer 1 Report
Introduction is adequate, mathodology and results clearly presented. Discussion should be improved. It could be more detailed and focused also on previous research describing mechanisms of the effects of detraining on the presented results. In discussion it should be mentioned that some results are also results of the increased body weight (CMJ). Presented results of LA thresholds are not supporting text in disscusion, esspecially in females sample.
Author Response
Dear Reviewer,
Thank you very much for your time and valuable comments, which all have been considered and incorporated. The detailed list of responses is given below. We hope that the modifications and explanation will be acceptable for you.
Yours sincerely,
Rydzik, corresponding author
Comments/Suggestions 1:
Introduction is adequate, mathodology and results clearly presented. Discussion should be improved. It could be more detailed and focused also on previous research describing mechanisms of the effects of detraining on the presented results. In discussion it should be mentioned that some results are also results of the increased body weight (CMJ). Presented results of LA thresholds are not supporting text in disscusion, esspecially in females sample.
Response 1:
Dear Sir/Madam,
We are very grateful for your opinion about the suitability of the manuscript attached. Thank you very much for the time and effort in reviewing this manuscript and for all your comments and suggestions, which we find very relevant and useful. All the changes have been marked in blue in the attached revised version of the manuscript.
In line with the comments, we have corrected some of the discussion on the results of LA threshold, this has been clarified in line with the comments. We also marked the role of body weight in the results of Countermovement Jump, which has a significant impact on the results obtained in both watts and centimeters.
We also understand the remark regarding references to other works, but due to the specific tests we have performed and the duration of the detraining period, we are not able to refer to more other works in the discussion, because our knowledge does not indicate that someone had previously made measurements on such conditions as we.
Reviewer 2 Report
I enjoyed the paper. This is a very interesting topic; however, it has a critical drawback in the theoretical background and unclear methods. My decision will make the authors disappoint, but I hope they are helpful for improving their paper.
Line 59.
The authors mentioned that examining the effects of a 12-week transition period is interesting. But why is it interesting? They need to provide theoretical background or arguments on why it is interesting to examine the effects of a 12-week transition 59 period more than other periods. I believe the above is the most important and only originality of this paper and no theoretical argument by the authors caused my understanding of no value of this paper. The authors must express the above point in the manuscript.
Line 64-69.
These parts indicate the purposes of this research. However, it is very hard to understand the purposes described. The sentence should include all key variables (e.g., IV, DV, moderating variable) which are easily understood among readers.
Line 73.
The authors mentioned "due to various circumstances". They need to specify what circumstances happened and what caused 5 participants to be excluded from your samples.
Line 76.
What does "at the second measurement" mean? Is this "a test after the 12-week transition period"? The authors need to define each test before and after the 12-week transition period and explain how each test was operated in the manuscript.
Author Response
Dear Reviewer,
Thank you very much for your time and valuable comments, which all have been considered and incorporated. The detailed list of responses is given below. We hope that the modifications and explanation will be acceptable for you.
Yours sincerely,
Rydzik, corresponding author
Dear Sir/Madam,
We are very grateful for your opinion about the suitability of the manuscript attached. Thank you very much for the time and effort in reviewing this manuscript and for all your comments and suggestions, which we find very relevant and useful. Below you will find a point-by-point reply to all your comments and suggestions. All the changes have been marked in green in the attached revised version of the manuscript.
Comments/Questions/Suggestions 1:
I enjoyed the paper. This is a very interesting topic; however, it has a critical drawback in the theoretical background and unclear methods. My decision will make the authors disappoint, but I hope they are helpful for improving their paper.
Line 59.
The authors mentioned that examining the effects of a 12-week transition period is interesting. But why is it interesting? They need to provide theoretical background or arguments on why it is interesting to examine the effects of a 12-week transition 59 period more than other periods. I believe the above is the most important and only originality of this paper and no theoretical argument by the authors caused my understanding of no value of this paper. The authors must express the above point in the manuscript.
Response 1:
Thank you for your valuable attention. The text has been corrected as recommended. We argued the legitimacy of our research. We paid attention to the duration of the detraining period in relation to the starting calendars of swimmers at a high national level in Poland and in the world. We also referred to other works dealing with similar topics, thus explaining the legitimacy and originality of the duration of the detraining period in our work.
Comments/Questions/Suggestions 2:
Line 64-69.
These parts indicate the purposes of this research. However, it is very hard to understand the purposes described. The sentence should include all key variables (e.g., IV, DV, moderating variable) which are easily understood among readers.
Response 2:
We agree with the expert reviewer and the manuscript has been revised in line with the reviewers' suggestions. We have clarified all the variables, specifying how the data were collected during the tests in water. We hope that these changes will have a positive effect on the clarity of the text for the readers.
Comments/Questions/Suggestions 3:
Line 73.
The authors mentioned "due to various circumstances". They need to specify what circumstances happened and what caused 5 participants to be excluded from your samples.
Response 3:
We agree with this opinion, the text has been improved. We clarified the reasons for which 5 players were excluded from our research.
Comments/Questions/Suggestions 4:
Line 76.
What does "at the second measurement" mean? Is this "a test after the 12-week transition period"? The authors need to define each test before and after the 12-week transition period and explain how each test was operated in the manuscript.
Response 4:
We agree with the reviewer’s comment and the text has been corrected. We have clarified and unified all descriptions of measurement periods in the text.
Reviewer 3 Report
Manuscript ijerph - 1641447
Effects of a 12-week detraining period on physical capacity, 2 power, and speed in elite swimmers
Thank you very much for giving me the opportunity to learn about the research described in this article. In my opinion, the obtained results are very valuable not only for highly qualified athletes, but also for people who practice health-oriented activities. Knowing the effects of the rest period with regard to the body's metabolism allows you to take the right steps when it comes to resuming intensive activity in planning your training process.
The article is written in accordance with good practice of writing scientific articles and in my opinion meets the requirements of the journal. The authors correctly introduce the reader to the issues described (Introduction section), where they correctly formulate the objectives of the study and, what is most important, realize and describe them in the article.
All other parts of the paper are also correctly structured. Research methods, methods of statistical analysis and discussion of results also meet the requirements of a scientific article.
However, analysing the content of the article it seems that some content should be improved, clarified or supplemented. These shortcomings make it difficult to read, sometimes to understand, but they do not affect the substantive quality of the content.
COMMENTS
Abstract
15 line – there is in the text '... at 12-month intervals...' while there is in the title '... a 12-week detraining...'; please clarify.
17 line – there are abbreviations - CMJ, ACMJ - which should be explained.
Materials and Methods - Body weight and composition
95-99 lines - this part describes the method of body composition measurement using electrical bioimpedance. It is generally known that this method requires appropriate preparation of the examined person for the analysis (a day without alcohol, emptying the bladder, dry whole body without any metal elements, as for women - time without menstruation, etc.). In the case of subjects who have not fulfilled these requirements, the results obtained will be falsified. In my opinion, it would be appropriate to supplement this content with an assurance that the subjects were informed and fulfilled the recommendation. It would also be good to include some literature item containing the exact procedure.
98-99 lines - the authors use the unit of mass expression "Kg" (also elsewhere in the article). The correct notation is "kg".
Results
215 line – The authors give a parameter e.g. "... Gender factor for Body weight...". It cannot be found anywhere how it was counted and for what purpose the authors wanted to use it. Please clarify this.
219 line (and in many other places) - the abbreviation "ES" appears, which is not explained. I assume it means "effect size". Am I right? Please explain this even earlier in the "Statistical Analysis" section.
Table 1.
All abbreviations used in the table should be explained below the one. Some names or numerical values are in bold, others are not. Is this intentional? If so, please explain why.
Table 2.
Same comments as for Table 1. Additionally, the authors use the notation "m/s", which is incorrect according to current rules. It should be or m·s-1.
Discussion
276-279 lines – The authors cite Ormsbee and Arciero, but do not provide a literature reference from References, that is [20]. Please complete.
280 line – The same observation applies to Zacca et al. - [11].
References
The authors refer to outdated scientific content. Out of 35 literature items only 3 from recent years are from 2019 - [11], 2018 - [30] and 2020 - [35]. If this was intentional please clarify and if not please update the literature. It is very important.
Author Response
Dear Reviewer,
Thank you very much for your time and valuable comments, which all have been considered and incorporated. The detailed list of responses is given below. We hope that the modifications and explanation will be acceptable for you.
Yours sincerely,
Rydzik, corresponding author
Dear Sir/Madam,
We are very grateful for your opinion about the suitability of the manuscript attached. Thank you very much for the time and effort in reviewing this manuscript and for all your comments and suggestions, which we find very relevant and useful. Below you will find a point-by-point reply to all your comments and suggestions. All the changes have been marked in yellow in the attached revised version of the manuscript.
Comments/Questions/Suggestions 1:
Abstract
15 line – there is in the text '... at 12-month intervals...' while there is in the title '... a 12-week detraining...'; please clarify.
Response 1:
Thank you for your attention, there was a mistake that has been corrected in the text.
Comments/Questions/Suggestions 2:
17 line – there are abbreviations - CMJ, ACMJ - which should be explained.
Response 2:
We agree with the expert reviewer and the manuscript was revised follow the reviewers suggestions.
Comments/Questions/Suggestions 3:
Materials and Methods - Body weight and composition
95-99 lines - this part describes the method of body composition measurement using electrical bioimpedance. It is generally known that this method requires appropriate preparation of the examined person for the analysis (a day without alcohol, emptying the bladder, dry whole body without any metal elements, as for women - time without menstruation, etc.). In the case of subjects who have not fulfilled these requirements, the results obtained will be falsified. In my opinion, it would be appropriate to supplement this content with an assurance that the subjects were informed and fulfilled the recommendation. It would also be good to include some literature item containing the exact procedure.
Response 3:
The text has been amended in line with this note. We have clarified the conditions that were followed during the measurements. We have also added an additional item of literature describing the method of body composition measurement using electrical bioimpedance.
Comments/Questions/Suggestions 4:
98-99 lines - the authors use the unit of mass expression "Kg" (also elsewhere in the article). The correct notation is "kg".
Response 4:
The authors agree with the reviewer’s comment and the text has been corrected.
Comments/Questions/Suggestions 5:
Results
215 line – The authors give a parameter e.g. "... Gender factor for Body weight...". It cannot be found anywhere how it was counted and for what purpose the authors wanted to use it. Please clarify this.
219 line (and in many other places) - the abbreviation "ES" appears, which is not explained. I assume it means "effect size". Am I right? Please explain this even earlier in the "Statistical Analysis" section.
Response 5:
Thank you for your attention. In the section on the statistical description it was explained what the main factors were used in the analysis of variance and it was explained that the abbreviation "ES" refers to the size of the effect – effect size.
Comments/Questions/Suggestions 6:
Table 1.
All abbreviations used in the table should be explained below the one. Some names or numerical values are in bold, others are not. Is this intentional? If so, please explain why.
Table 2.
Same comments as for Table 1. Additionally, the authors use the notation "m/s", which is incorrect according to current rules. It should be or m·s-1.
Response 6:
All comments regarding both tables have been applied in the article. We standardized the font, we added translations of all abbreviations and improved the notation of speed units. We agree that the table now looks clearer to the readers.
Comments/Questions/Suggestions 7:
Discussion
276-279 lines – The authors cite Ormsbee and Arciero, but do not provide a literature reference from References, that is [20]. Please complete.
280 line – The same observation applies to Zacca et al. - [11].
Response 7:
Thank you for this remark, we have corrected all related errors in the text.
Comments/Questions/Suggestions 9:
References
The authors refer to outdated scientific content. Out of 35 literature items only 3 from recent years are from 2019 - [11], 2018 - [30] and 2020 - [35]. If this was intentional please clarify and if not please update the literature. It is very important.
Response 9:
Thanks for your suggestion. The text has been updated with an additional 4 items from 2009-2019. The rest of the bibliography is based on scientific research from earlier years due to the specificity of research on the period of de-training and on the still up-to-date news from the world of swimming.
Round 2
Reviewer 3 Report
Manuscript ijerph-1641447
Effects of a 12-week detraining period on physical capacity, 2 power, and speed in elite swimmers
Dear Authors,
Thank you very much for taking my comments so seriously and adapting the text to them. I very much appreciate it. However, I am concerned about the outdated literature. Your answers and correction of the paper satisfy me and I will request that your paper be accepted for publication. As for the literature, I leave the decision to the editorial office.